# Adsorption of Cu(II) and Zn(II) Ions from Aqueous Solution by Gel/PVA-Modified Super-Paramagnetic Iron Oxide Nanoparticles

**DOI:** 10.3390/molecules23112982

**Published:** 2018-11-15

**Authors:** Anudari Dolgormaa, Chang-jiang Lv, Yin Li, Jian Yang, Jun-xing Yang, Peng Chen, Hong-peng Wang, Jun Huang

**Affiliations:** 1Zhejiang Provincial Key Lab for Chemical and Biological Processing Technology of Farm Product, School of Biological and Chemical Engineering, Zhejiang University of Science and Technology, Hangzhou 310023, China; anu_d3@outlook.com (A.D.); yangtzelv@zju.edu.cn (C.-j.L.); cherryli1986@126.com (Y.L.); cp1030090003@163.com (P.C.); 2State Key Laboratory Breeding Base of Dao-di Herbs, National Resource Center for Chinese Material Medical, China Academy of Chinese Medical Sciences, Beijing 100700, China; yangchem2012@163.com; 3Institute of Geographical Sciences and Natural Resources Research, Chinese Academy of Sciences, Beijing 100101, China; yangajx@igsnrr.ac.cn

**Keywords:** super-paramagnetic iron oxide nanoparticles, gelatin, PVA, adsorption, heavy metal

## Abstract

Super-paramagnetic iron oxide nanoparticles (SPIONs)/gelatin (gel)/polyvinyl alcohol (PVA) nanoparticles were designed and synthesized by the co-precipitation method and further modified with gel and PVA. These nanoparticles were used for the removal of Cu(II) and Zn(II) from aqueous solutions. The adsorbents were rich in different functional groups for chemisorption and showed effective adsorption properties. The adsorption of Cu(II) and Zn(II) on the SPIONs/gel and SPIONs/gel/PVA materials were investigated with respect to pH, adsorption kinetics, and adsorption isotherms. The adsorption data was fitted to the Langmuir, Freundlich, and Sips models at the optimum pH 5.2 (±0.2) over 60 min; SPIONs/gel showed maximum adsorption capacities of 47.594 mg/g and 40.559 mg/g for Cu(II) and Zn(II); SPIONs/gel/PVA showed those of 56.051 mg/g and 40.865 mg/g, respectively. The experimental data fitted the pseudo-second-order model, indicating that the process followed chemical monolayer adsorption. In addition, the SPIONs/gel/PVA showed better stability and Cu(II) adsorption efficiency than SPIONs/gel.

## 1. Introduction

In the past decade, toxic heavy metal ion pollution has had the harmful effects on the environment. Heavy metal ions such as Pb, Cu, Ni, Zn, Cd, and Cr are toxic to humans, animals, and plants even at low concentrations [1]. The accumulation of Cu(II) in living organisms can cause liver and brain damage, heart disease, skin conditions, and pancreas issues [2]. Zn(II) poisoning in humans causes nausea, dizziness, and dehydration [3]. Many technologies have been developed for the removal of heavy metal ions from water, such as chemical precipitation, electrochemical reduction, ion exchange, membrane separation, solvent extraction, coagulation, and adsorption [4]. However, the technologies have their own disadvantages as known as ineffective at lower heavy metal ions concentration, are not economical, and have specific working conditions [5]. Among these technologies, the adsorption process is attractive because of its low cost and high efficiency. Widely used adsorbents include carbonaceous materials, clays, zeolites, composite materials, biomass, nanomaterials, and polymeric materials [6,7,8]. The efficiency of adsorbents depends on their specific surface areas and their chemical stabilities [9]. Therefore, the choice of suitable materials for the adsorption of heavy metal ions from wastewater is very important.

The use of nanoparticle materials offers many advantages because of their small sizes and interesting physical properties. The use of magnetic nanoparticles is greatly interesting, especially in wastewater treatment. Magnetic sorbents could be easily separated by an external magnetic field from the liquid phase, which would assist in removing organic and inorganic pollutants using these materials [10]. The adsorptive removal of heavy metals, including those in aqueous solutions, by magnetic nanoparticles has been reported [8,11,12,13].

Gelatin is a natural water-soluble biopolymer with a variety of advantages such as environmental friendliness, biodegradability, biocompatibility, and low cost. It is feasible to use gelatin as an adsorbent material for various ionic species because of its abundant –OH, –NH_2_, and –COOH groups [14,15]. Therefore, gelatin coated on the surfaces of magnetic nanoparticles could enhance their adsorption capacity for heavy metal ions through the reactivity of its own functional groups [16,17,18]. Magnetic nanoparticles modified by gelatin have been reported in use for scavenging U(VI) under a series of environmental conditions [14]. However, the drawbacks of gelatin include its poor mechanical properties and rapid degradation in wet conditions [19,20,21]. The blending of gelatin with other polymers is one effective strategy to improve the physical performance [22], such as adsorption efficiency and reusability.

Poly (vinyl alcohol) (PVA) is a widely used polymer with good biocompatibility, non-toxicity, high mechanical strength, thermal stability, pH stability, and low cost. Composite materials of gelatin and PVA have good adsorption capacities for heavy metals [23,24]. Hui et al. [6] prepared PVA/gelatin hydrogel beads for Pb(II) removal, and the maximum adsorption capacity of Pb(II) was 211.86 mg/g and the adsorption capacity remained stable in four sequential adsorption–desorption cycles. Hence, coating PVA on the surface of gelatin-modified magnetic nanoparticles might reduce the aggregation of magnetic nanoparticles and provide good adsorption properties for heavy metals. A SPIONs/gelatin composite material has already been synthesized for drug delivery, chemotherapy, and the cleansing of radionuclide-bearing effluents, but not for the removal of Cu(II) and Zn(II). Moreover, SPIONs/gelatin and SPIONs/gelatin/PVA composite materials are reported for the first time here in the removal of heavy metal ions from aqueous solution.

In this work, gelatin/PVA-modified super-paramagnetic iron oxide nanoparticles (SPIONs) were prepared by the co-precipitation of Fe(II) and Fe(III) ions in an ammonia solution, and then treated with gelatin and PVA, characterized, and evaluated for adsorption of Cu(II) and Zn(II) from aqueous solutions. The objective of this work was to explore the feasibility of SPIONs/gelatin (SPIONs/gel) and SPIONs/gelatin/poly (vinyl alcohol) (SPIONs/gel/PVA) composites and to use them for removing Cu(II) and Zn(II) from aqueous solutions.

## 2. Results and Discussion

### 2.1. Characterization of the Adsorbents

The surface functional groups on the modified magnetic nanoparticles were determined by FTIR analysis; the spectra are shown in Figure 1. The Fe–O bond stretching at 566–575 cm^−1^ can be found in all samples [25,26]. Gelatin has functional groups such as –OH and –NH groups (between 3200–3500 cm^−1^), –CH_2_ group (asymmetric C–H stretching vibration 2946 cm^−1^ and symmetric C–H stretching vibration 2870 cm^−1^), amide group (1648 cm^−1^ and 1534 cm^−1^) [14,16,27]. PVA has functional groups such as –OH group (3200–3500 cm^−1^), –CH_2_ groups (2866 cm^−1^), C–O–C groups (1090 cm^−1^ stretching vibration) [6,7,28].

The FTIR results of gelatin, PVA, SPIONs/gel, and SPIONs/gel/PVA showed the carboxyl and amide groups at the range of 1632–1648 cm^−1^ and hydroxyl group at 1335–1396 cm^−1^. Moreover, as can be seen from FTIR results of SPIONs/gel and SPIONs/gel/PVA, the PVA and gelatin layers cannot be observed easily because the gelatin and PVA showed similar functional groups in their spectra, and small signals of the amide bands and C–O stretching vibrations were observed from the SPIONs/gel and SPIONs/gel/PVA. These functional groups are favorable for the adsorption of heavy metal ions.

The TEM images of SPIONs (a), SPIONs/gel (b), and SPIONs/gel/PVA (c) are shown in Figure 2. The TEM images showed that SPIONs, SPIONs/gel, and SPIONs/gel/PVA particles are quasi-spherical in shape with rough surfaces. The sizes of SPIONs, SPIONs/gel, and SPIONs/gel/PVA were measured to be approximately 14.02 nm, 22.63 nm, and 18.6 nm, respectively, from their TEM images. Furthermore, the DLS measurement showed the sizes of SPIONs, SPIONs/gel, and SPIONs/gel/PVA were with 15.2 nm, 22.47 nm, and 17.74 nm, respectively. The size of the magnetic nanoparticles results from TEM and DLS were identical.

Figure 3 showed XRD diffraction patterns of SPIONs, SPIONs/gel, and SPIONs/gel/PVA.

The diffraction pattern for SPIONs had eight broad peaks at 31.249°, 36.820°, 38.524°, 44.762°, 55.622°, 59.303°, 65.185°, and 77.249°, corresponding to the (220), (311), (222), (400), (422), (511), (440), and (533) planes, respectively [7,29]. All of the observed diffraction peaks were indexed by the structure of the Fe_3_O_4_. Meanwhile, the peaks of SPIONs/gel and SPIONs/gel/PVA showed a highly crystalline nature.

The magnetization values of the pure SPIONs/gel and SPIONs/gel/PVA were 56.382 and 57.765 emu/g, respectively. Figure 4 showed that the magnetization properties of these two materials were very similar, the modified layers should not influence the magnetization efficiency of the nanoparticles, since the SPIONs/gel and SPIONs/gel/PVA have very thin modified layers of gelatin and PVA. This results are consistent with the TGA results of the nanoparticles. It is clear that SPIONs/gel and SPIONs/gel/PVA samples can be easily separated from aqueous solution by applying an external magnetic field [14,30].

The TGA curves of SPIONs, SPIONs/gel, and SPIONs/gel/PVA at the temperature range 50–800 °C are shown in Figure 5. The samples were subjected to heating to 800 °C at a heating rate of 10 °C/min. The Figure 5 showed that the mass of the materials was reduced to some extent at the temperature range of 50–200 °C, indicating the removal of adsorbed water and some oxygen-containing functional groups at this temperature range. At the first stage, the mass losses of SPIONs/gel and SPIONs/gel/PVA at temperatures below 150 °C were 0.75% and 1.92%, respectively. In the second stage, when the temperature was between 150 and 250 °C, the mass losses of SPIONs/gel and SPIONs/gel/PVA were 0.87% and 2.58%, respectively. In the third stage, when the temperature was between 250 and 470 °C, the mass losses of the SPIONs/gel and SPIONs/gel/PVA were 9.85% and 9.42%, respectively. It was observed that the SPION/gel had a total weight loss of approximately 11.47% and SPIONs/gel/PVA had a total weight loss of approximately 13.92%. Therefore, the weight fraction of gelatin in the SPIONs/gel composite is approximately 10.72% and the weight fraction of PVA in the SPIONs/gel/PVA composite is approximately 1.25%, which suggested the thickness of organic layer of SPIONs/gel and SPIONs/gel/PVA are very thin, and both particles have similar thickness of organic layer.

### 2.2. Adsorption

#### 2.2.1. Effect of pH

The effects of pH on the SPIONs/gel and SPIONs/gel/PVA are shown in Figure 6. As can be seen from Figure 6 with the increase of the pH of the aqueous solution, the adsorption capacity for Cu(II) and Zn(II) on both adsorbents is increased. At low pH, this may relate to the higher concentration and mobility of H^+^ ions. Meanwhile, minimal adsorption is favored by the high solubility and ionization of metallic salts in the acidic medium [31]. The suitable pH for Cu(II) and Zn(II) removal for all adsorbents is determined as 5–6, at which the maximum adsorption capacity occurs for both adsorbents. Moreover, the precipitation of heavy metal hydroxides occurs for a pH enhanced further [32]. If the pH is more than 6, the heavy metal ions will be precipitated from the aqueous solution [33]. Accordingly, the initial pH of the solutions was 5.2 (±0.2) for both solutions in the following adsorption experiments.

#### 2.2.2. Adsorption Kinetics

The effects of contact time on the kinetics of Cu(II) and Zn(II) adsorption by the SPIONs/gel and SPIONs/gel/PVA adsorbents are displayed in Figure 7. As can be seen in Figure 7, the removal of Cu(II) and Zn(II) on the SPIONs/gel and SPIONs/gel/PVA is rapid in the initial 5 min, and the removal of Cu(II) increases sharply in the initial 30 min before continuing at a slower rate and finally reaching equilibrium at 60 min. Therefore, 60 min was chosen as the optimum contact time for the adsorption experiments. The adsorption rate is faster than those shown by some reported magnetic adsorbent materials [34,35,36]. Keochaiyom et al. [4] reported that the adsorption equilibrium time was 24 h for Zn(II), Cd(II), Pb(II) on magnetic chlorapatite nanoparticles. Ghasemi et al. [36] reported that the adsorption process reached equilibrium within 90 min for the removal of Zn(II) from an aqueous solution.

In order to investigate the mechanisms of adsorption of Cu(II) and Zn(II) on the SPIONs/gel and SPIONs/gel/PVA, fitting was determined according to the pseudo-first-order and pseudo-second-order kinetic models. The sorption kinetic parameters including *K*_1_, *K*_2_, *Q_t_*, and the correlation coefficients *R*^2^ are listed in Table 1. As can be seen, the correlation coefficients *R*^2^ for the pseudo-second-order model are higher than those for the pseudo-first-order model. The calculated *Q_t_* values are close to the experimental *Q_e_* values with the pseudo-second-order model for both adsorbents. These results suggest that the sorption kinetics of heavy metal ions on SPIONs/gel and SPIONs/gel/PVA can be described by the pseudo-second-order model, which means that the adsorption rate depends on chemical sorption [37]. For illustration, functional groups such as carboxyl, amino, and hydroxyl groups [27,28,38] on the surfaces of both adsorbents, as characterized by the FTIR results, were active in heavy metal ion binding on the SPIONs/gel and SPIONs/gel/PVA. Similar results have been reported by Vo et al. [28] and Wang et al. [38] for the removal of heavy metal ions by other magnetic adsorbents.

#### 2.2.3. Adsorption Isotherms

The adsorption isotherms of Cu(II) and Zn(II) on SPIONs/gel and SPIONs/gel/PVA at pH 5 are presented Figure 8. It can be seen from Figure 8 that the adsorption capacities of Cu(II) and Zn(II) on both adsorbents are increased with increasing initial concentrations of the metal ions. The SPIONs/gel showed the equilibrium adsorption capacities of 20.1213 mg/g and 14.9092 mg/g for Cu(II) and Zn(II); SPIONs/gel/PVA showed those of 13.7879 mg/g and 9.7803 mg/g, respectively, at the equilibrium concentration of 0.1 mg/mL and 25 °C. The adsorption capacity of Cu(II) on bare iron oxide was just 0.15 mg/g, much less than that on another two adsorbents, so that the following adsorption experiments were carried on SPIONs/gel and SPIONs/gel/PVA. Moreover, the adsorption capacities of Cu(II) on SPIONs/gel and SPIONs/gel/PVA are much more than those of Zn(II); adsorption on SPIONs/gel in particular shows a higher adsorption capacity than SPIONs/gel/PVA.

The experimental data points were analyzed by the Langmuir, Freundlich, and Sips models.

The calculated parameters together with the correlation coefficients for the Langmuir, Freundlich, and Sips models are shown in Table 2. As seen in Table 2, the adsorption of Cu(II) and Zn(II) on both adsorbents is well correlated (*R*^2^ < 0.99) with the Langmuir model, Freundlich model, and Sips model. All models show good fits with the experimental data. The Langmuir model describes monolayer adsorption on the homogenous surface of the adsorbents [29]. The dimensionless separation parameter *R_L_* is the essential characteristic of the Langmuir isotherm model and expressed by:(1)$$R_{L}=1/\left(1+K_{L}C_{0}\right)$$
where *C*_0_ is initial concentration of the metal ions (mg/mL) and the definition of *K_L_* is as explained for Equation (4).

The *R_L_* value is classified as irreversible (*R_L_* = 0), linear (*R_L_* = 1), favorable (0 < *R_L_* < 1), and unfavorable (*R_L_* > 1) [39,40]. In this work, all the values of *R_L_* are between 0.1305 and 0.2274, indicating that the experimental data fall between zero and 1, which is an indication of the favorable adsorption of the two metal ions on the adsorbents.

The Freundlich model describes the surface heterogeneity of the sorbent, indicating multilayer adsorption. The 1/*n* value from the Freundlich model between 0.5 and 1.0, represents favorable adsorption; on the other hand, a value of 1/*n* > 1 suggests weak adsorption bonds between the adsorbent and adsorbate, indicating unfavorable adsorption [41]. The 1/*n* values from the Freundlich model for the adsorption of SPIONs/gel were 0.3164 and 0.3367 for Cu(II) and Zn(II); SPIONs/gel/PVA were those of 0.4373 and 0.4399, respectively. These values are lower than 1, indicating that the adsorption of Cu(II) and Zn(II) is favorable on the both adsorbents.

The Sips model is the combination of Langmuir and Freundlich model and explains adsorption on both homogeneous and heterogeneous surfaces. Its behavior is similar to Freundlich model; it reduces to the Freundlich isotherm at a low adsorbate concentration and to the Langmuir model at a high adsorbate concentration [42,43]. The adsorption of Cu(II) and Zn(II) on both adsorbents appears to be a monolayer adsorption process, because the Langmuir and Sips models both fit better than the Freundlich model. The correlation coefficients *R*^2^ from the Langmuir and Sips models are nearly 1, suggesting that the adsorption of Cu(II) and Zn(II) on both adsorbents appears to be a monolayer adsorption process. SPIONs/gel showed the maximum adsorption capacities of 47.594 mg/g and 40.559 mg/g for Cu(II) and Zn(II); SPIONs/gel/PVA showed those of 56.051 mg/g and 40.865 mg/g, respectively. Meanwhile, the interaction of carboxyl groups in both adsorbents with Cu(II) and Zn(II) formed bidentate chelates, a chemical sorption process [6].

The maximum adsorption capacities of both adsorbents were higher than the equilibrium adsorption capacities and SPIONs/gel/PVA showed a higher adsorption capacity than SPIONs/gel, indicating that the adsorption properties were improved significantly by PVA. These results suggest that the SPIONs/gel/PVA adsorbent showed monolayer adsorption for low concentrations of metal ions, while multilayer adsorption processes occur for increased concentrations. The SPIONs/gel/PVA has more hydroxyl groups on the surface than SPIONs/gel and greater rigidity, which increased the maximum adsorption capacities for both ions. The maximum adsorption capacity of Cu(II) is higher than that of Zn(II), meaning that the chelating effect is more suitable for Cu(II). Meanwhile, Cu(II) and Zn(II) have different characteristic properties, such as the difference in covalency index ($X_{m}^{2}$r, where $X_{m}$ is electronegativity and r is ionic radius). Zhu et al. [44] recorded that the $X_{m}^{2}$r is a measure for a metal ions of the importance of covalent interactions relative to that of ionic interactions. The covalency index increased in the following order: Zn (2.04) < Cu (2.64), suggesting that Cu(II) has a stronger attraction that Zn(II), which agreed with the report by Nieboer et al. [45] in which metal ions preferentially interacted with functional groups in the following order: S- > N- > O-containing groups. This indicates that larger values of $X_{m}^{2}$r, correspond to more characteristics of soft acids (HSAB theory). The adsorption capacities of SPIONs/gel and SPIONs/gel/PVA for Cu(II) and (Zn) from aqueous solution were compared with those reported in previously published works. The adsorption capabilities of SPIONs/gel and SPIONs/gel/PVA for Cu(II) and Zn(II) from aqueous solution were higher than those of some magnetic nanomaterials such as poly(N-2-aminoethylacrylamide) magnetite nanoparticles and magnetic nanoparticle-decorated tea waste [46,47].

#### 2.2.4. Regeneration

Repeated use of adsorbents and the recovery of the adsorbed metal ions are important parameters indicating economic efficiency. In this study, the regeneration of the SPIONs/gel and SPIONs/gel/PVA was tested by consecutive adsorption–desorption processes performed three times using the same adsorbents.

The adsorption capacity of Cu(II) on SPIONs/gel and SPIONs/gel/PVA were decreased by 5.05% and 7.28%. The adsorption capacity of Zn(II) on SPIONs/gel and SPIONs/gel/PVA were decreased by 7.87% and 12.57%, after three cycles, respectively. The reusability of these materials were compared with some magnetic adsorbent materials, which was used to similar preparation of iron oxide nanoparticles, initial concentration of heavy metal ions and cycling usage conditions. The adsorption capacity of Cu(II) on the Mg/Fe layered double hydroxide loaded with Magnetic (Fe_3_O_4_) carbon spheres [48], and the magnetic Fe_3_O_4_/carbon nanotube (CNT) [49] were decreased by 18.6%, and 17.6%, after three cycles of usage, respectively. The adsorption capacity of Cu(II) and Zn(II) on magnetic nanoparticles decorated tea waste (Fe_3_O_4_-tea) was decreased by 9.5% and 17.1% after six cycles [46]. As shown in Figure 9, after three cycles that the SPIONs/gel and SPIONs/gel/PVA exhibit good regeneration performance and can support long-term usage in the removal of Cu(II) and Zn(II).

## 3. Experimental

### 3.1. Materials

Ferric chloride (FeCl_3_), ferrous chloride (FeCl_2_·4H_2_O), ammonia solution (NH_3_·H_2_O), gelatin, polyvinyl alcohol (PVA), copper sulfate pentahydrate (CuSO_4_·5H_2_O), zinc sulfate heptahydrate (ZnSO_4_·7H_2_O), hydrogen chloride (HCl), nitric acid (HNO_3_), sodium hydroxide (NaOH), and all test solutions were prepared using deionized water. All chemicals were of were analytical grade and used without further purification.

### 3.2. Synthesis of SPIONs/Gel/PVA

Firstly, SPIONs were prepared by co-precipitating Fe(II) (FeCl_2_·4H_2_O) and Fe(III) (FeCl_3_) ions in ammonia solutions, as described in the literature [50,51]. 5.4066 g of FeCl_3_ and 3.3 g FeCl_2_·4H_2_O were dissolved in 100 mL deionized water and then heated to 80 °C under mechanical stirring in Ar atmospheric conditions. Afterwards, 15 mL of 29% NH_3_·H_2_O solution was added to the solution of iron salts, which formed a black precipitate. The mixture was continually stirred at 80 °C for 1 h, cooled to 70 °C, and stirred for 1 h. After that, 30 mL of gelatin solution (5% *w*/*w*) was added to the mixture and the synthesis was continued for 1 h. Then 30 mL of PVA solution (10% *w*/*w*) was added to the mixture. The synthesis was continued for another 1 h, and the solid phase was collected and washed several times with deionized water and ethanol. The resulting powder was dried at room temperature (25 °C) in vacuum (0.1 MPa) for 24 h. Pure SPIONs and SPIONs/gel were synthesized in a similar condition without the addition of the PVA solution.

### 3.3. Characterization of Adsorbent

The morphologies of SPIONs, SPIONs/gel, and SPIONs/gel/PVA were characterized using transmission electron microscopy (TEM, JEOL JEM-2100, Tokyo, Japan). The sizes of the SPIONs, SPIONs/gel, and SPIONs/gel/PVA were measured by Dynamic Light Scattering (DLS). All DLS measurements were performed with a Malvern Zetasizer Nano ZS particle analyzer (ZEN3600, Malvern, United Kingdom) at wavelength of 633 nm and He-Ne laser at scattering angle of 173° at 25 °C. In this analysis, 1 mL of particle suspension (each sample was diluted to approximately 0.1 mg/mL) was employed and placed in 10 mm × 10 mm quartz cuvette. The intensity and average diameters were calculated by the Zetasizer Nano Software 7.01 (Malvern, UK). The X-ray diffraction (XRD) patterns of SPIONs, SPIONs/gel, and SPIONs/gel/PVA were determined using an X-ray diffractometer using Cu Kα (λ = 1.54 Å, scanning rate of 5° min^−1^ in the range of 10–80° at 40 kV and 20 mA, Rigaku, RINT2000). The Fourier-transform infrared (FTIR) spectra of SPIONs, SPIONs/gel, and SPIONs/gel/PVA were recorded by using an FTIR spectrophotometer (Vertex 70, Brucker Optik GmbH, Karlsruhe, Germany) in the range 400–4500 cm^−1^. The fine powder of KBr was mixed with the sample (0.1–0.5%) and then ground to get a homogenous mixture. The homogenous mixture was compressed under 10 MPa pressure to a small disk for analysis (Transmission ~10%). The data of the sample from FTIR were plotted by Origin Pro 9.1 software. Thermogravimetric analysis (TGA) was performed by using a STA 449 F3 DSC/DTA-TG analyzer (Netzsch Germany, Wunsiedel, Germany), with the temperature heating range from 50 to 900 °C under N_2_ atmosphere, and the heating rate of 10 °C/min.

### 3.4. Adsorption

The adsorption experiments of Cu(II) and Zn(II) were performed via the batch adsorption method. Metal ion solutions used in the adsorption experiments were prepared by dissolving salts (CuSO_4_·6H_2_O and ZnSO_4_·7H_2_O) in deionized water. The initial and final heavy metal ion concentrations were measured by a flame atomic absorption spectrophotometer (AAS, AA6300, Shimadzu Corporation, Kyoto, Japan).

#### 3.4.1. Effect of pH

25 mg of the adsorbents were added to 25 mL of 0.1 mg/mL Cu(II) and Zn(II) solution at a given initial concentration in a 50 mL flask. The pH of the contact solution was adjusted from 2 to 6 using 0.1 mol/L HCl and 0.1 mol/L NaOH solutions. The flask was then placed in an automatic shaker at 25 °C and shaken at 130 rpm for 60 min. The adsorbents were separated by an external magnetic field and the residual heavy metal concentrations were measured by AAS. All the experiments were conducted in duplicate and each data point was measured three times. The adsorption capacities of the SPIONs/gel and SPIONs/gel/PVA were calculated by the following equation [52]:(2)$$Q_{e}=\left(C_{0}-C_{e}\right)\times V/m$$
where *Q_e_* is the adsorption capacity of the SPIONs/gel or the SPIONs/gel/PVA (mg/g). *C*_0_ and *C_e_* are the initial and final concentrations of the metal ions (mg/mL), respectively. *V* is the volume of the solution (mL), and *m* is the weight of the adsorbents (g).

#### 3.4.2. Adsorption Kinetics

The effect of contact time on adsorption was determined at different time intervals from 5 to 240 min. 25 mL of Cu(II) and Zn(II) with the initial concentration of 0.1 mg/mL was mixed with 0.025 g of SPIONs/gel and SPIONs/gel/PVA adsorbents under shaking at 130 rpm and 25 °C for different contact times. The adsorbents were then separated and the solution concentrations were measured. The data obtained from these experiments were fitted to the pseudo-first-order and pseudo-second-order kinetic models [29,53].

The equations of these models are given by:

Pseudo-first-order model:(3)$$Q_{t}=Q_{e}\left(1-e^{-K_{1}t}\right)$$

Pseudo-second-order model:(4)$$Q_{t}=tQ_{e}{}^{2}K_{2}/\left(tK_{2}Q_{e}+1\right)$$
where *Q_t_* (mg/g) is the amount of metal ions adsorbed by SPIONs/gel or SPIONs/gel/PVA at time *t*, and *K*_1_ and *K*_2_ are the rate constants of the pseudo-first-order (min^−1^) and pseudo-second-order (min^−1^) models, respectively.

#### 3.4.3. Adsorption Isotherms

25 mL of Cu(II) and Zn(II) metal ions solutions with different initial concentrations of 0.1, 0.2, 0.3, 0.4, 0.5, 0.75 and 1 mg/L was mixed with 0.025 g of adsorbents and shaken at 25 °C and 130 rpm for 60 min. After that, the adsorbents were separated by an external magnetic field and the residual concentrations of heavy metal ions were measured by AAS. The experimental data points were analyzed by the Langmuir, Freundlich, and Sips models.

Langmuir equation:(5)$$C_{e}/Q_{e}=1/\left(K_{L}Q_{max}+C_{e}/Q_{max}\right)$$

Freundlich equation:(6)$$Q_{e}=K_{F}C_{e}^{1/n}$$

Sips equation:(7)$$Q_{e}=Q_{max}K_{s}C_{e}^{1/n}/(1+K_{s}C_{e}^{1/n})$$
where the definition of *Q_e_* is the same as that for equation 1, *Q_max_* is the maximum adsorption capacity of the adsorbent (mg/g), *K_L_* is the Langmuir constant related to the adsorption strength or intensity (mg/g), *K_F_* is the Freundlich constant, *K_S_* is the Sips model constant (mL/mg)^1/*n*^, and 1/*n* is the heterogeneity factor.

#### 3.4.4. Regeneration

25 mg of the SPIONs/gel and SPIONs/gel/PVA were added to 25 mL of 0.1 mg/L Cu(II) and Zn(II) and shaken at 130 rpm and 25 °C for 60 min. The SPIONs/gel and SPIONs/gel/PVA adsorbents were separated and the residual concentration was measured. The collected adsorbents were mixed with 0.1 mol/L HNO_3_ solution (25 mL) and shaken at 130 rpm and 25 °C for 60 min. After that, the adsorbents were collected and washed repeatedly with deionized water to neutralize the acidic condition and recycled three times.

## 4. Conclusions

In this work, magnetic nanoparticles were modified by gelatin and PVA for the removal of Cu(II) and Zn(II). Copper and Zinc are very important metals for industries, electroplating, paint, pigments, wood, metal plating, and so on. These enriched acidic solutions can be concentrated and continue used to produce their salts and materials. The SPIONs/gel and SPIONs/gel/PVA were used for the first time for the removal of heavy metal ions from aqueous solutions. SPIONs/gel and SPIONs/gel/PVA can adsorb the two metal ions effectively; SPIONs/gel showed equilibrium adsorption capacities of 18.067 mg/g and 14.143 mg/g for Cu(II) and Zn(II); SPIONs/gel/PVA showed those of 12.904 mg/g and 9.303 mg/g, respectively, at 0.1 mg/mL and 25 °C. The equilibrium data fitted very well with the pseudo-second-order model, indicating chemical adsorption by the chelating effect. The SPIONs/gel/PVA adsorbent is more stable than that of SPIONs/gel, with greater adsorption efficiency for Cu(II).

## Figures and Tables

**Figure 1 molecules-23-02982-f001:**
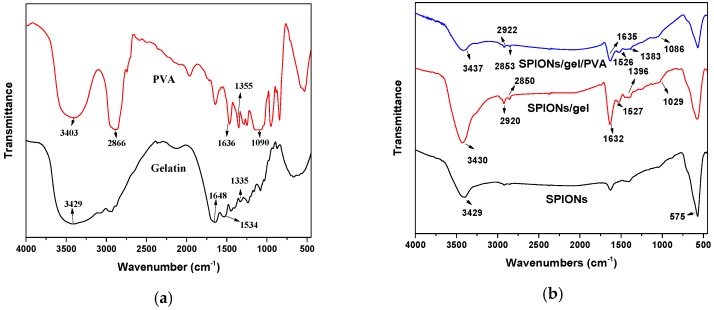
The FTIR spectra of (**a**) poly vinyl acetate (PVA), gelatin (gel), (**b**) Super-paramagnetic iron oxide nanoparticles (SPIONs), SPIONs/gel and SPIONs/gel/PVA.

**Figure 2 molecules-23-02982-f002:**
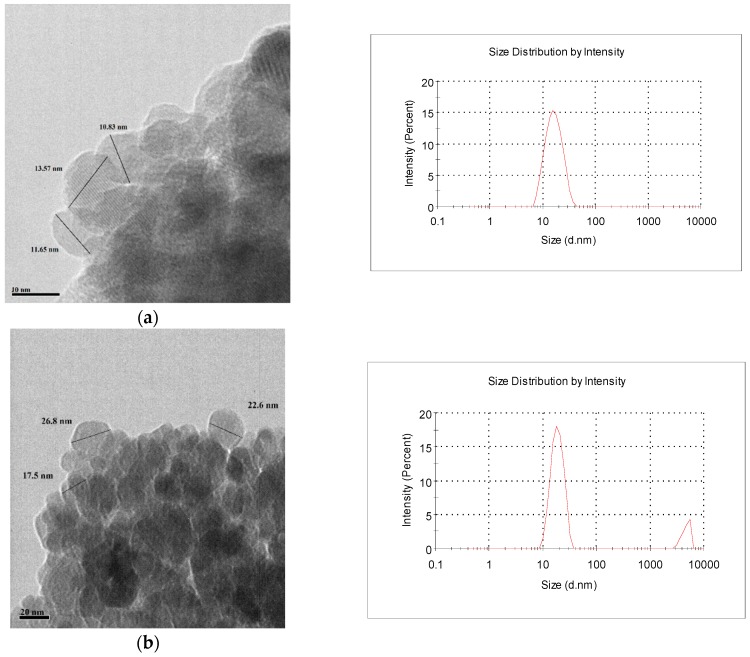
The TEM images and Dynamic Light Scattering (DLS) of (**a**) SPIONs, (**b**) SPIONs/gel, and (**c**) SPIONs/gel/PVA.

**Figure 3 molecules-23-02982-f003:**
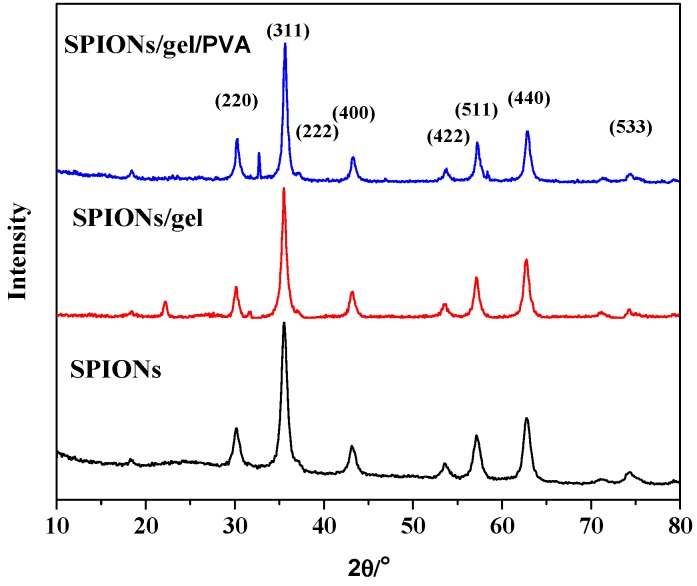
XRD spectra of SPIONs, SPIONs/gel and SPIONs/gel/PVA.

**Figure 4 molecules-23-02982-f004:**
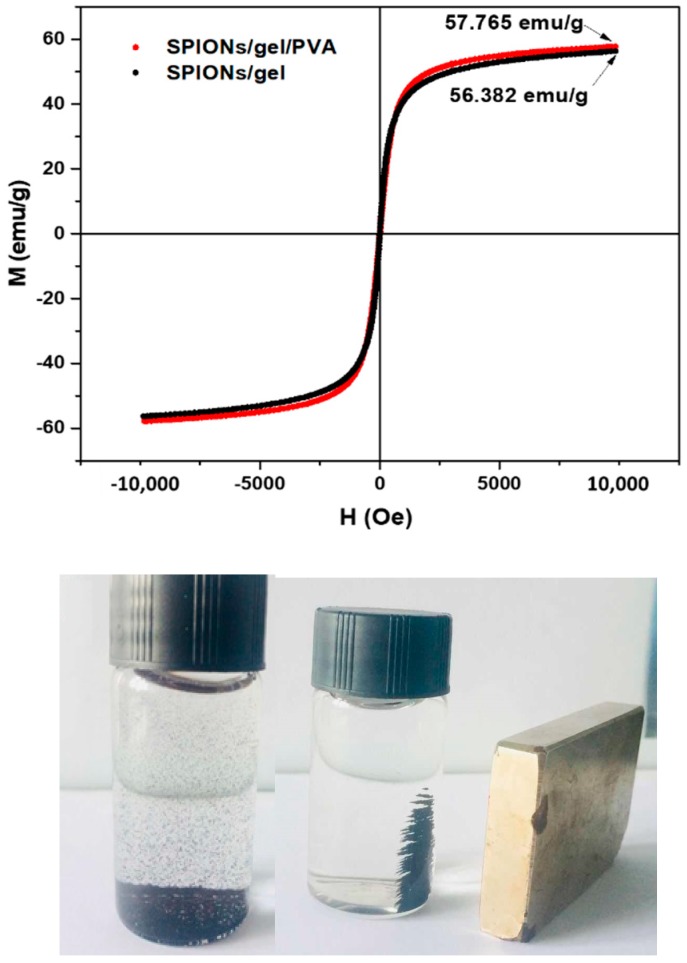
Magnetization curve of SPIONs/gel and SPIONs/gel/PVA; the magnetization responds to the external magnetic field of magnetic adsorbents.

**Figure 5 molecules-23-02982-f005:**
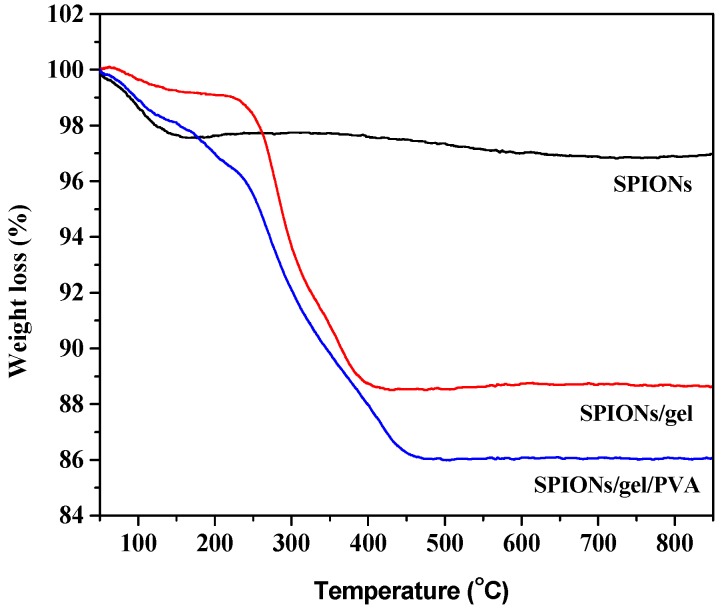
Thermographs of SPIONs, SPIONs/gel and SPIONs/gel/PVA.

**Figure 6 molecules-23-02982-f006:**
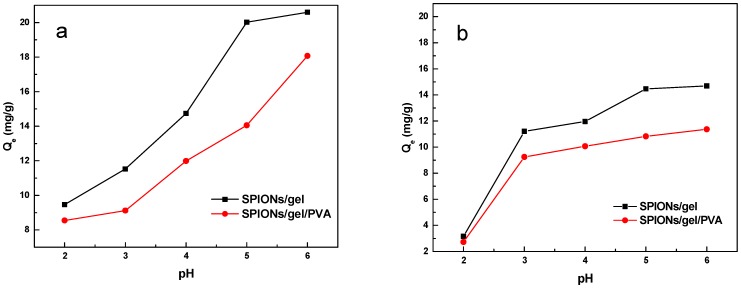
Effect of pH on Cu(II) (**a**) and Zn(II) (**b**) ions adsorption by SPIONs/gel and SPIONs/gel/PVA.

**Figure 7 molecules-23-02982-f007:**
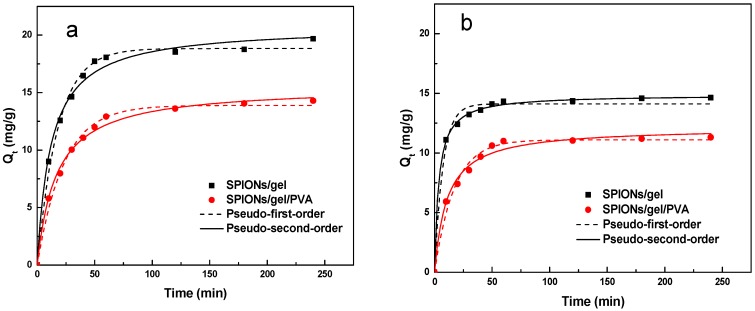
Adsorption kinetics for the adsorption of Cu(II) (**a**) and Zn(II) (**b**) by SPIONs/gel and SPIONs/gel/PVA.

**Figure 8 molecules-23-02982-f008:**
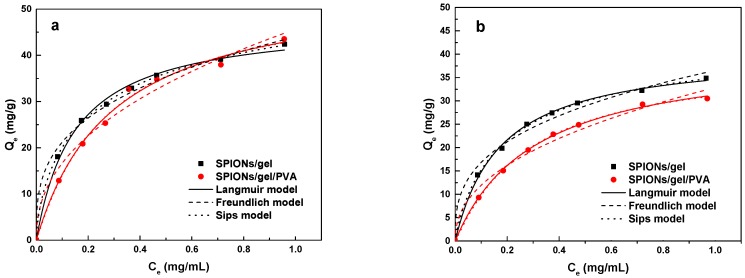
Equilibrium adsorption isotherm of Cu(II) (**a**) and Zn(II) (**b**) by the SPIONs/gel and SPIONs/gel/PVA.

**Figure 9 molecules-23-02982-f009:**
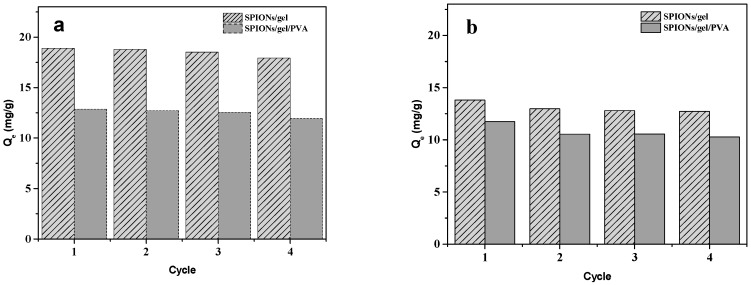
Regeneration of SPIONs/gel and SPIONs/gel/PVA for Cu(II) (**a**) and Zn(II) (**b**).

**Table 1 molecules-23-02982-t001:** Adsorption kinetic parameters for adsorption of Cu(II) and Zn(II) on SPIONs/gel and SPIONs/gel/PVA.

Metal	Adsorbents	Pseudo-First-Order			Pseudo-Second-Order		
		*Q_t_* (mg/g)	*K*_1_ (1/min)	*R* ^2^	*Q_t_* (mg/g)	*K*_2_ (g/(mg·min))	*R* ^2^
Cu(II)	SPIONs/gel	18.834	0.055	0.992	20.779	0.004	0.991
	SPIONs/gel/PVA	13.886	0.043	0.992	15.55	0.003	0.994
Zn(II)	SPIONs/gel	14.109	0.137	0.986	14.888	0.019	0.998
	SPIONs/gel/PVA	11.101	0.059	0.983	12.146	0.008	0.986

**Table 2 molecules-23-02982-t002:** Fitted result of the adsorption isotherms of the Cu(II) and Zn(II) by SPIONs/gel and SPIONs/gel/PVA.

Isotherm	Parameters	Cu(II)		Zn(II)	
		SPIONs/Gel	SPIONs/Gel/PVA	SPIONs/Gel	SPIONs/Gel/PVA
Langmuir model	*Q_max_* (mg/g)	47.594	56.051	40.559	40.865
	*K_L_* (mL/mg)	6.66	3.397	5.727	3.267
	*R* ^2^	0.995	0.993	0.997	0.999
Freundlich model	*K_F_* ((mg/g)(mL/mg)1/*n*)	43.991	45.658	36.515	32.866
	1/*n*	0.3164	0.4373	0.3367	0.4399
	*R* ^2^	0.993	0.976	0.986	0.979
Sips model	*Q_max_* (mg/g)	60.422	54.823	43.974	38.976
	*K_S_* (mL/mg)1/*n*	2.382	3.713	3.854	3.961
	1/*n*	0.6816	1.0319	0.8632	1.071
	*R* ^2^	0.999	0.991	0.998	0.999

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
