# Peer review of "Adsorption of Cu(II) and Zn(II) Ions from Aqueous Solution by Gel/PVA-Modified Super-Paramagnetic Iron Oxide Nanoparticles"

_molecules, 2018, doi:10.3390/molecules23112982_

Round 1

Reviewer 1 Report

The manuscript describes the synthesis of gelatine-iron oxide nanoparticle-compounds and .PVA/gelatine-iron oxide nanoparticle compounds for the adsorption of heavy metal ions. The authors describe the synthesis and characterize the materials with TEM, XRD, STA, magnetometry and IR spectroscopy. The characterized materials are tested towards their properties to act as adsorbents for divalent copper and zinc ions. Therefore adsorption kinetics, adsorption isotherms as well as the influence of pH and reusability of the materials are thoroughly discussed.

All in all, the manuscript is quite interesting and the investigation focuses on application of the materials. The references are balanced and well placed. Especially the introduction is interesting, introduces the problem and includes the state of the art. However, there are a few points which need improvement. Here is a conclusion of my essential issues: The characterizations and their descriptions, the discussion of the material properties need improvement and the discussion should be reshaped a bit.

Issues point-by-point:

Nomenclature of your particles: there is no gamma Fe3O4 and I would not recommend to call your particles magnetite/Fe3O4 in general but refer to iron oxide nanoparticles. I will refer to this issue especially in the XRD section.

Figure 1/IR discussion:

Your Y-axis is labelled strangely. Do you really have 4% Transmittance for PVA, 3 for gelatin and so on or is the axis labelled wrongly? I think the spectra are normalized. Please refer to the normalization and relable the axis.

Why is neither gelatin nor PVA visible in the spectra of the “modified” samples?

You discuss in the text about OH, NH, CH2/3 and C=O stretches. However, the OH vibration is visible on bare nanoparticles as well as on the modified particles. Even the almost not visible peaks closely below 3000 cm-1 are similar on all nanoparticle materials. The same is true for the band above 1600 cm-1 might derive from physisorbed water and is therefore similar on all particles (Maybe more prominent on PVA particles). There is no band around 1700-1750 cm-1 (at least not visible for my eyes) as described by the authors. Well there is the possibility of chemisorbed carbonic acids on the surface as well which would fit this range and can occur during synthesis or washing of nanoparticles (CO2 dissolution in basic aqueous solution and adsorption of derived carbonic acid).  The most prominent amide band around 1650 cm-1, which is visible for the gelatin reference, is not visible in the modified samples. Furthermore, no C-O vibration around 1000-1200 cm-1 is visible in the spectrum of modified particles while the PVA reference illustrates them quite pronounced.

Figure 2/TEM discussion:

What means most of the particles are approximately 20 nm? Did you measure and count them? Is it the mean value of this distribution? Then please add the number of counted particles in your experimental section. If you did not count them, then you should assess the particle diameter by measuring the particles from the pictures.

Why do you compare the particle size to the particles reported by Jia et al.? They used a completely different synthesis with stabilization of iron oxide nanoparticles and without gelatin. However, in order to assess the size of your modification layer, it would be nice to compare the size with your bare nanoparticles. You should add a picture of the bare particles and compare sizes. Is it possible to visualize the modification layer of gelatine or PVA in the TEM pictures? Is it visible?

Figure 3/XRD:

Why are the diffractograms of PVA/gelatin-nanoparticles and gelatin-nanoparticles completely identical? I mean even the noise is identical which is really strange and beyond belief.

Furthermore, there are two reflections around 20° and 24° which can be assigned (210) and (211) planes of maghemite. Thus, I would recommend not to use Fe3O4 for your nomenclature.

Figure 4/ Magnetization curves:

Please provide the measured data points instead of two lines. Why is the saturation magnetization of gelatin/PVA-particles larger than the saturation magnetization of gelatin particles? From the XRD, you have the same crystalline phase and from STA you have a higher iron oxide content in gelatin particles. Can you explain this discrepancy?

Figure 5/ Thermographs:

Why is there a mass increase of the bare nanoparticles? Under which atmosphere did you conduct your measurements? It is not possible to understand or discuss the measurements without knowing these conditions. Air or inert gas behave complete differently concerning the ashing behaviour of gelatin and PVA. Furthermore, how did you pretreat your sample? Is it possible that different amounts of water are bound to different particles? Especially the region below 200°C may be assigned to physisorbed water.

This whole characterization section needs to be rewritten.

For the adsorption section it would be quite interesting to compare the adsorption behaviour of modified particles with bare particles. Especially as there is no real evidence that the modification is located at the nanoparticle surface. This might either confirm the nice adsorption qualities of the modifications or demonstrate that the ion adsorption is through the iron oxide surface. Therefore I would recommend the pH studies, adsorption kinetics and adsorption isotherms to be conducted with bare iron oxide nanoparticles.

Another question: Why did you not consider pH 7 for your measurements? Your binding capacity might even increase for most ions. There is no proof of a maximum adsorption capacity for any ion on any nanoparticle used here. The use of pH 5.2 seems arbitrary. Can you justify this pH for further experiments?

Can you explain the following sentence?

When the initial concentration of Cu(II) is low, the adsorption rate of Fe3O4/gel/PVA is increased like that of the Fe3O4/gel, but when the initial concentration is increased from 0.3 mg/mL to 0.5 mg/mL, the adsorption capabilities are rapidly increased.”

I do not understand the sentence. Why should the adsorption rate be increased when the initial concentration is low? The adsorption rate is independent of the concentration of the adsorbed ion. The adsorption capability is also not altered by the ion concentration. This sentence should be erased completely.

Adsorption isotherms: What is the reason to set the final starting concentration of ions to 05 mg/mL? Your models describe much higher maximum binding capacities. You should at least measure until the maximum binding capacity in order to accurately describe the binding behavior with adsorption isotherms.

I am a great fan of discussion to different adsorption models but I do not think the different models should be described in the discussion to such an extent. The equations and the descriptions/explanations of the models should be included in the experimental part. Only the outcome of the fits should be included in the discussion.

Regeneration:

The following sentence should be accompanied by a comparing reference from the literature in order to proof your regeneration performance:

As shown in Figure 9, after three cycles that the Fe3O4/gel and Fe3O4/gel/PVA exhibit good regeneration performance and can support long-term usage in the removal of Cu(II) and Zn(II).”

Experimental part:

Mention the X-ray source of your XRD (I assume Cu Kalpha) and what kind of measurement you performed.

What FTIR method did you use? (Transmission?) How was it prepared (KBr%)? How was it normalized or further manipulated?

How did you pretreat your STA samples and under which atmosphere did you conduct the measurements?

Conclusion:

Why do you mention the equilibrium adsorption capacities originating from 0.1 g/L ions? Is there a reason? Then state it. Furthermore, this conclusion is not conclusive at all. It does not conclude the whole discussion. Just conclude your findings.

Author Response

Response to Reviewer’s Comments on Molecules-379597

“Adsorption of Cu(II) and Zn(II) ions from aqueous solution by gel/PVA-modified magnetic nanoparticles”

By: Anudari Dolgormaa1, Chang-jiang Lv1, Yin Li1, Jian Yang2, Jun-xing Yang3, Peng Chen1, Hong-peng Wang1,*, Jun Huang1,*

We appreciate very much the constrictive and encouraging comments made by all the reviewers. This manuscript was revised by considering all comments and suggestions carefully. Detailed responses to each comment and a summary of revisions made are given below.

Reviewer 1#

The manuscript describes the synthesis of gelatine-iron oxide nanoparticle-compounds and PVA/gelatine-iron oxide nanoparticle compounds for the adsorption of heavy metal ions. The authors describe the synthesis and characterize the materials with TEM, XRD, STA, magnetometry and IR spectroscopy. The characterized materials are tested towards their properties to act as adsorbents for divalent copper and zinc ions. Therefore, adsorption kinetics, adsorption isotherms as well as the influence of pH and reusability of the materials are thoroughly discussed.

All in all, the manuscript is quite interesting and the investigation focuses on application of the materials. The references are balanced and well placed. Especially the introduction is interesting, introduces the problem and includes the state of the art. However, there are a few points which need improvement. Here is a conclusion of my essential issues: The characterizations and their descriptions, the discussion of the material properties need improvement and the discussion should be reshaped a bit.

1.                  Nomenclature of your particles: there is no gamma Fe3O4 and I would not recommend to call your particles magnetite/Fe3O4 in general but refer to iron oxide nanoparticles. I will refer to this issue especially in the XRD section.

Action/response: The symbol “ɣ” was erased from Section 2 and Section 3.

2.                  Figure 1/IR discussion: Your Y-axis is labelled strangely. Do you really have 4% Transmittance for PVA, 3 for gelatin and so on or is the axis labelled wrongly? I think the spectra are normalized. Please refer to the normalization and relable the axis.

Why is neither gelatin nor PVA visible in the spectra of the “modified” samples?

You discuss in the text about OH, NH, CH2/3 and C=O stretches. However, the OH vibration is visible on bare nanoparticles as well as on the modified particles. Even the almost not visible peaks closely below 3000 cm-1 are similar on all nanoparticle materials. The same is true for the band above 1600 cm-1 might derive from physisorbed water and is therefore similar on all particles (Maybe more prominent on PVA particles). There is no band around 1700-1750 cm-1 (at least not visible for my eyes) as described by the authors. Well there is the possibility of chemisorbed carbonic acids on the surface as well which would fit this range and can occur during synthesis or washing of nanoparticles (CO2 dissolution in basic aqueous solution and adsorption of derived carbonic acid). The most prominent amide band around 1650 cm-1, which is visible for the gelatin reference, is not visible in the modified samples. Furthermore, no C-O vibration around 1000-1200 cm-1 is visible in the spectrum of modified particles while the PVA reference illustrates them quite pronounced.

Action/response: Section 2, Page 3: The spectra of PVA and gelatin raw materials could not be clearly displayed within Figure 1. We separated Figure 1 into Figure 1 (a) and (b) as follows:

(a)

(b)

Section 2, Page 3: The title of “Figure 1” has been replaced as follows: The FTIR spectra of (a) PVA, gelatin, (b) Fe3O4, Fe3O4/gel, and Fe3O4/gel/PVA

Section 2. Paragraph 1: Result and discussion of FTIR analysis has been revised as follows:

The surface functional groups on the modified magnetic nanoparticles were determined by FTIR analysis; the spectra are shown in Figure 1. The Fe–O bond stretching at 566–575 cm−1 can be found in all samples [25, 26]. Gelatin has functional groups such as -OH and -NH groups (between 3200-3500 cm-1), -CH2 group (asymmetric C-H stretching vibration 2946 cm-1 and symmetric C-H stretching vibration 2870 cm-1), amide group (1648 cm-1 and 1534 cm-1) [14, 16, 27]. PVA has functional groups such as -OH group (3200-3500 cm-1 and 1355 cm-1), -CH2 groups (2866 cm-1), C-O-C groups (1090 cm-1 stretching vibration) [6, 7, 28]. The FTIR results of gelatin, PVA, Fe3O4/gel, and Fe3O4/gel/PVA showed the carboxyl and amide groups at the range of 1632-1648 cm-1 and hydroxyl group at 1335-1396 cm-1. These functional groups are favorable for the adsorption of heavy metal ions.

[6]        Hui B.; Zhang Y.; Ye L. Structure of PVA/gelatin hydrogel beads and adsorption mechanism for advanced Pb(II) removal, J. Ind. Eng. Chem. 2015, 21, 868-876.

[7]        Lv L.; Chen N.; Feng C.; Gao Y.; Li M. Xanthate-modified magnetic chitosan/poly (vinyl alcohol) adsorbent: Preparation, characterization, and performance of Pb(II) removal from aqueous solution, J. Taiwan Instit. Chem. Eng. 2017, 78, 485-492

[14]      Tang P.; Shen J.; Hu Z.; Bai G.; Wang M.; Peng B.; Shen R.; Linghu W.; High-efficient scavenging of U(VI) by magnetic Fe3O4@gelatin composite, J. Mol. Liq. 2016, 221, 497-506.

[16]      Rajzer I.; Menaszek E.; Kwiatkowski R.; Planell J.A.; Castano O. Electrospun gelatin/poly(epsilon-caprolactone) fibrous scaffold modified with calcium phosphate for bone tissue engineering, Mater. Sci. Eng. C. Mater. Biol. Appl. 2014, 44, 183-190.

[25]      Nithya K.; Sathish A.; Kumar P.S.; Ramachandran T. Fast kinetics and high adsorption capacity of green extract capped superparamagnetic iron oxide nanoparticles for the adsorption of Ni(II) ions, J. Ind. Eng. Chem. 2018, 59, 230-241.

[26]      Yang L.; Tian J.; Meng J.; Zhao R.; Ma J.; Jin T. Modification and Characterization of Fe3O4 Nanoparticles for Use in Adsorption of Alkaloids. Molecules, 2018, 23(3), 562

[27]      Dil N.N.; Sadeghi M. Free radical synthesis of nanosilver/gelatin-poly (acrylic acid) nanocomposite hydrogels employed for antibacterial activity and removal of Cu(II) metal ions, J. Hazard. Mater. 2018, 351, 38-53.

[28]      Vo T.K.; Park H.K.; Nam C.W.; Kim S.D.; Kim J. Facile synthesis and characterization of γ-AlOOH/PVA composite granules for Cr(VI) adsorption, J. Ind. Eng Chem. 2018, 60, 485-492.

3.                  Figure 2/TEM discussion: What means most of the particles are approximately 20 nm? Did you measure and count them? Is it the mean value of this distribution? Then please add the number of counted particles in your experimental section. If you did not count them, then you should assess the particle diameter by measuring the particles from the pictures.

Action/response: The sizes of Fe3O4/gel and Fe3O4/gel/PVA were measured by Nano Measurer 1.2 software from their TEM images. The sizes of the Fe3O4, Fe3O4/gel, and Fe3O4/gel/PVA were further confirmed by Dynamic Light Scattering (DLS) using Zetasizer Nano (ZEN 3600, Malvern, United Kingdom). Figure 2 has been replaced with the modified TEM images in the revised manuscript, additionally, the particle size distributions obtained from Dynamic Light Scattering were added in the revised manuscript. Related text and figures were revised as follows:

Section 2, Paragraph 2.2: The TEM images of Fe3O4 (a), Fe3O4/gel (b) and Fe3O4/gel/PVA (c) are shown in Figure 2. The TEM images showed that Fe3O4, Fe3O4/gel and Fe3O4/gel/PVA particles are quasi-spherical in shape with rough surfaces. The sizes of Fe3O4, Fe3O4/gel and Fe3O4/gel/PVA were measured to be approximately 14.02 nm, 22.63 nm, and 18.6 nm, respectively from their TEM images. Furthermore, the DLS measurement showed the sizes of Fe3O4, Fe3O4/gel and Fe3O4/gel/PVA were with 15.2 nm, 22.47 nm, and 17.74 nm, respectively. The size of the magnetic nanoparticles results from TEM and DLS were identical.

Section 3 page 13: “The sizes of the Fe3O4, Fe3O4/gel, and Fe3O4/gel/PVA were measured by Dynamic Light Scattering (DLS) using Zetasizer Nano (ZEN 3600, Malvern, United Kingdom). “was added into the characterization of adsorbent.

(a)

(b)

(c)

Fig. 2 The TEM images and DLS spectra of Fe3O4 (a), Fe3O4/gel (b) and Fe3O4/gel/PVA (c)

4.                  Why do you compare the particle size to the particles reported by Jia et al.? They used a completely different synthesis with stabilization of iron oxide nanoparticles and without gelatin. However, in order to assess the size of your modification layer, it would be nice to compare the size with your bare nanoparticles. You should add a picture of the bare particles and compare sizes. Is it possible to visualize the modification layer of gelatine or PVA in the TEM pictures? Is it visible?

Action/response:

Section 2, Page 3: In our study, the size of the bare magnetic nanoparticle were around 10-16 nm, and similar results have been reported by Jia et al. Furthermore, this comparison is not important, so the reference [29] was removed.

The modified layer was not able to be determinable in TEM, since these layers are very thin. Similar results have been reported, such as chitosan layer on iron oxide was also not shown in TEM images [1, Shalaby et al]. Furthermore, the TGA curves of Fe3O4/gel and Fe3O4/gel/PVA have approved that the gelatin and gelatin/PVA were modified on the surface of the magnetic nanoparticles.

Literatures:

[1]        Shalaby Th. I., Fikrt N. M., Mohamed M. M., El Kady M. F. Preparation and characterization of iron oxide nanoparticles coated with chitosan for removal of Cd(II) and Cr(VI) from aqueous solution, J. Water Sci. Tech. 2014, 70, 1004-1010.

5.                  Figure 3/XRD: Why are the diffractograms of PVA/gelatin-nanoparticles and gelatin-nanoparticles completely identical? I mean even the noise is identical which is really strange and beyond belief. Furthermore, there are two reflections around 20° and 24° which can be assigned (210) and (211) planes of maghemite. Thus, I would recommend not to use Fe3O4 for your nomenclature.

Action/response: The Fe3O4/gel and Fe3O4/gel/PVA were measured again by X-ray diffractometer using Cu Kα (λ=1.54 Å, scanning rate of 5° min-1 in the range of 10-80° at 40 kV and 20 mA, Rigaku, RINT2000) and the original Figure 3 has been updated. The XRD result showed strong six characteristic peaks of Fe3O4, and just two weak signals of (210) and (211) for maghemite, these results are similar to reported results in literatures Kulkarni et al [1] and Dong et al [2], the nomenclature of “Fe3O4will be kept.

Literatures:

[1]        Kulkarni S. A., Sawadh P. S., Kokate K. K., Synthesis and characterization of Fe3O4 nanoparticles for engineering applications, 2012, 2, 17-18”

[2]        Dong Y., Yang Z., Sheng Q., Zheng J., Solvothermal synthesis of Ag@Fe3O4 nanosphere and its application as hydrazine sensor, 2018, 538, 371-377.

Section 2, the Figure 3. XRD spectra of Fe3O4, Fe3O4/gel and Fe3O4/gel/PVA was revised as:

Figure 3. XRD spectra of Fe3O4, Fe3O4/gel and Fe3O4/gel/PVA

6.                  Figure 4/ Magnetization curves: Please provide the measured data points instead of two lines. Why is the saturation magnetization of gelatin/PVA-particles larger than the saturation magnetization of gelatin particles? From the XRD, you have the same crystalline phase and from STA you have a higher iron oxide content in gelatin particles. Can you explain this discrepancy?

Action/response: As can be seen from Figure 4, the magnetization values of Fe3O4/gel and Fe3O4/gel/PVA were very close to each other. The difference of the magnetization values was 1.38 emu/g, it is very little and can be ignored.

Section 2. Paragraph 2.4: The following sentences have been added:

The Figure 4 showed that the magnetization properties of these two materials were very similar, the modified layers should not influence the magnetization efficiency of the nanoparticles, since the Fe3O4/gel and Fe3O4/gel/PVA have very thin modified layers of gelatin and PVA. This results are consistent with the TGA results of the nanoparticles.

7.                  Figure 5/ Thermographs: Why is there a mass increase of the bare nanoparticles? Under which atmosphere did you conduct your measurements? It is not possible to understand or discuss the measurements without knowing these conditions. Air or inert gas behave complete differently concerning the ashing behaviour of gelatin and PVA. Furthermore, how did you pretreat your sample? Is it possible that different amounts of water are bound to different particles? Especially the region below 200°C may be assigned to physisorbed water.

Action/response: The conditions for TGA test was added into the section 3.

Thermogravimetric analysis (TGA) was performed by using a STA 449 F3 DSC/DTA-TG analyzer (Netzsch Germany), with the temperature heating range from 50 to 900 °C under N2 atmosphere, and the heating rate of 10 °C/min.

Section 2., page 6. The Figure 5 showed that the mass of the materials was reduced to some extent at the temperature range of 50–200 °C, indicating the removal of adsorbed water and some oxygen-containing functional groups at this temperature range.

Section 2., page 6. Line 124-125: It was observed that the Fe3O4/gel had a total weight loss of approximately 11.47% and Fe3O4/gel/PVA had a total weight loss of approximately 13.92%.

Section 2., page 6. Line 133: -which suggested the thickness of the organic layers of Fe3O4/gel and Fe3O4/gel/PVA is very thin and both particles have similar thickness of organic layers.

Section 2, Figure 5. The TGA graphs of Fe3O4, Fe3O4/gel, and Fe3O4/gel/PVA has been revised as:

8.                  For the adsorption section it would be quite interesting to compare the adsorption behaviour of modified particles with bare particles. Especially as there is no real evidence that the modification is located at the nanoparticle surface. This might either confirm the nice adsorption qualities of the modifications or demonstrate that the ion adsorption is through the iron oxide surface. Therefore, I would recommend the pH studies, adsorption kinetics and adsorption isotherms to be conducted with bare iron oxide nanoparticles.

Action/response: Adsorption capacity of the bare magnetic nanoparticle was tested at the same conditions with another two adsorbents. The adsorption capacity of Cu(II) on bare iron oxide was just 0.15 mg/g, much less than that on another two adsorbents (15-25 mg/g). Therefore, we did not continue the discussion of adsorption properties on bare magnetic nanoparticles for heavy metal ions.

The following sentences have been added into the revised manuscript to make this clearer:

The adsorption capacity of Cu(II) on bare iron oxide was just 0.15 mg/g, much less than that on another two adsorbents (15-25 mg/g), so that the following adsorption experiments were carried on Fe3O4/gel and Fe3O4/gel/PVA.

9.                  Another question: Why did you not consider pH 7 for your measurements? Your binding capacity might even increase for most ions. There is no proof of a maximum adsorption capacity for any ion on any nanoparticle used here. The use of pH 5.2 seems arbitrary. Can you justify this pH for further experiments?

Action/response: The following sentences and references have been added into the results of pH section:

If the pH more than 6, the heavy metal ions will be precipitated from the aqueous solution [33].

[33] Sutirman Z.A., Sanagi M. M., Karim K. J. A., Ibrahim W. A. W., Jume B. H., Equilibrium, kinetic and mechanism studies of Cu(II) and Cd(II) ions adsorption by modified chitosan bead, Int J Biol Macromol, 2018, 116, 255-263.

10.              Can you explain the following sentence? “When the initial concentration of Cu(II) is low, the adsorption rate of Fe3O4/gel/PVA is increased like that of the Fe3O4/gel, but when the initial concentration is increased from 0.3 mg/mL to 0.5 mg/mL, the adsorption capabilities are rapidly increased.” I do not understand the sentence. Why should the adsorption rate be increased when the initial concentration is low? The adsorption rate is independent of the concentration of the adsorbed ion. The adsorption capability is also not altered by the ion concentration. This sentence should be erased completely.

Action/response: Section 2, Page 7: “When the initial concentration of Cu(II) is low, the adsorption rate of Fe3O4/gel/PVA is increased like that of the Fe3O4/gel, but when the initial concentration is increased from 0.3 mg/mL to 0.5 mg/mL, the adsorption capabilities are rapidly increased.” this sentence has erased from result and discussion part.

11.              Adsorption isotherms: What is the reason to set the final starting concentration of ions to 0.5 mg/mL? Your models describe much higher maximum binding capacities. You should at least measure until the maximum binding capacity in order to accurately describe the binding behavior with adsorption isotherms. I am a great fan of discussion to different adsorption models but I do not think the different models should be described in the discussion to such an extent. The equations and the descriptions/explanations of the models should be included in the experimental part. Only the outcome of the fits should be included in the discussion.

Action/response: In this paper, the initial concentration of heavy metal ions for adsorption isotherm was set to be from 0.1 mg/mL to 0.5 mg/mL, since the industry wastewater contains a very low concentration of heavy metal ions. The maximum adsorption capacities are always higher than equilibrium capacities at any initial concentrations from the Langmuir model, it is an ideal calculated value.

The equations and the descriptions of the models were separated from the result and discussion part, included in the experimental part

12.              Regeneration: The following sentence should be accompanied by a comparing reference from the literature in order to proof your regeneration performance: As shown in Figure 9, after three cycles that the Fe3O4/gel and Fe3O4/gel/PVA exhibit good regeneration performance and can support long-term usage in the removal of Cu(II) and Zn(II).”

Action/response: These sentences and references have been added into result of regeneration section.

The adsorption capacity of Cu(II) on Fe3O4/gel and Fe3O4/gel/PVA decreased only 5.05% and 7.28% after three cycles, respectively, while the adsorption amount of Zn(II) on Fe3O4/gel and Fe3O4/gel/PVA decreased 7.87% and 12.57% after three cycles, respectively.

The adsorption capacity of Cu(II) on the Mg/Fe layered double hydroxide loaded with Magnetic(Fe3O4) carbon spheres after three times of cycles usage [48], and the magnetic Fe3O4/carbon nanotube (CNT) [49] decreased 18.6%, and 17.6%,.respectively. The adsorption capacity of Cu(II) and Zn(II) on magnetic nanoparticles decorated tea waste (Fe3O4-tea) decreased 9.5 % and 17.1 % after six cycles [46].

The regeneration of Fe3O4/gel and Fe3O4/gel/PVA showed better than another materials and this result suggest that Fe3O4/gel and Fe3O4/gel/PVA exhibit good regeneration performance and can support long term usage

These references were added into the manuscript.

[46]      Wen T.; Wang J.; Li X.; Huang S.; Chen Z.; Wang S.; Hayat T.; Alsaedi A.; Wang X. Production of a generic magnetic Fe3O4 nanoparticles decorated tea waste composites for highly efficient sorption of Cu(II) and Zn(II), JECE. 2017, 5(4), 3656-3666.

[48]      Xie Y., Yuan X., Wu Z., Zeng G., Jiang L., Peng X., Li H., Adsorption behavior and mechanism of Mg/Fe layered double hydroxide with Fe3O4-carbon spheres on the removal of Pb(II) and Cu(II), J Colloid Interface Sci,2018, 536, 440-455.

[49]      Yang Z. F., Li L. Y., Hsieh C. T., Juang R. S., Co-precipitation of magnetic Fe3O4 nanoparticles onto carbon nanotubes for removal of copper ions from aqueous solution. J. Taiwan Instit. Chem. Eng, 2018, 82, 56-63.

13.              Mention the X-ray source of your XRD (I assume Cu Kalpha) and what kind of measurement you performed.

Action/response: Page 10 line 300: The source of XRD and measurement performance added into the characterization of adsorbent.

The X-ray diffraction (XRD) patterns of Fe3O4, Fe3O4/gel, and Fe3O4/gel/PVA were determined using an X-ray diffractometer using Cu Kα (λ=1.54 Å, scanning rate of 5° min-1 in the range of 10-80° at 40 kV and 20 mA, Rigaku, RINT2000).

14.              What FTIR method did you use? (Transmission?) How was it prepared (KBr%)? How was it normalized or further manipulated?

Action/response: The small amount of powder samples (0.1-0.5 % of the KBr) mix with KBr powder and then were further manipulated. The transmission should be used just for the quantitative analysis, we need determine just the functional groups, so did not calculated here.

Section 3, page 13: The method KBr for FTIR spectroscopy added into the characterization of adsorbent.

The Fourier-transform infrared (FTIR) spectra of Fe3O4, Fe3O4/gel, and Fe3O4/gel/PVA were mixed with KBr and then recorded by using an FTIR spectrophotometer (Vertex 70, Brucker Optik GmbH, Germany) in the range 400–4500 cm−1 .

15.              How did you pretreat your STA samples and under which atmosphere did you conduct the measurements?

Action/response: 10 mg of each samples was placed in small crucibles and analyzed using a synchronous thermal analyzer. The heating rate was set at 10 °C/min, while the analyses were carried out in the temperature range of 50–800 °C under N2 atmosphere.

Section 3, page 13: The atmosphere of TGA analysis was added into the characterization of adsorbent.

Thermogravimetric analysis (TGA) was performed by using a STA 449 F3 DSC/DTA-TG analyzer (Netzsch Germany), with the temperature heating range from 50 to 900 °C under N2 atmosphere and the heating rate of 10 °C/min.

16.              Conclusion: Why do you mention the equilibrium adsorption capacities originating from 0.1 g/L ions? Is there a reason? Then state it. Furthermore, this conclusion is not conclusive at all. It does not conclude the whole discussion. Just conclude your findings.

Action/response: The largest removal efficiency of Cu(II) and Zn(II) on Fe3O4/gel and Fe3O4/gel/PVA was indicated when the initial concentration of heavy metal ions was 0.1 mg/mL. This result suggested that the adsorption of heavy metal ions on those adsorbents was efficient when the heavy metal ions were at low concentration.

Reviewer 2 Report

This paper describes the preparation and feasibility of gel/PVA-modified magnetic nanoparticles to eliminate Cu(II) and Zn(II) ions in wastewaters. I think that, in general, the work is appropiate and well presented. I have some considerations:

Introduction:

2nd paragraph. The authors indicated some technologies to removal of heavy metal ions from water (chemical precipitation, electrochemical reduction, ion exchange...). I suggest to include some disadvantages of these techniques.

Results and discussion. I think that the authors made a good characterization of the nanoparticles. 

I would like to ask if this method is adequate to removal the ions in large quantities of water, i. e.  applicable in a waste-water treatment plant and not only at laboratory scale. What is the "real" feasibility  also taking into account economical aspects?

Effect of pH. pag 5 line 136. It is not true that the maximum adsorption capacity was found at pH 6 because the authors did not show the results at pH 7. I am aware that at this pH, the precipitation of heavy metal hydroxide occurs. I just suggest to rewrite the text to clarify this aspect. 

Adsorption kinetics, pag 6, line 170. "...functional groups such as carboxyl, amino, and hydroxyl groups on the surfaces of both adsorbents, ..., were active in heavy metal ion bindin..." How is it  demonstrated?

Table 1. I don't think it would be necessary to include the sorption kinetic parameters corresponding to pseudo-first order since this model can not describe the results.

Experimental.

Regeneration. How do these acidic solution enriched with ions should be managed as a residue? Please, specify.

Author Response

Response to Reviewer’s Comments on Molecules-379597

“Adsorption of Cu(II) and Zn(II) ions from aqueous solution by gel/PVA-modified magnetic nanoparticles”

By: Anudari Dolgormaa1, Chang-jiang Lv1, Yin Li1, Jian Yang2, Jun-xing Yang3, Peng Chen1, Hong-peng Wang1,*, Jun Huang1,*

We appreciate very much the constrictive and encouraging comments made by all the reviewers. This manuscript was revised by considering all comments and suggestions carefully. Detailed responses to each comment and a summary of revisions made are given below.

Reviewer #2:

This paper describes the preparation and feasibility of gel/PVA-modified magnetic nanoparticles to eliminate Cu(II) and Zn(II) ions in wastewaters. I think that, in general, the work is appropiate and well presented. I have some considerations:

Introduction:

1.                  2nd paragraph. The authors indicated some technologies to removal of heavy metal ions from water (chemical precipitation, electrochemical reduction, ion exchange...). I suggest to include some disadvantages of these techniques.

Action/response: Page 1, line 39: The following sentences were added into the first paragraph in Introduction and marked red:

However, the technologies have own disadvantages as known as ineffective at lower heavy metal ions concentration, not economically, and specific working conditions [5].

And one reference was added into the manuscript.

[5] Bilal M., Shah J. A., Ashfaq T., Gardazi S. M., Tahir A. A., Pervez A., Haroon H., Mahmood Q. Waste biomass adsorbents for copper removal from industrial wastewater--a review, J Hazard Mater. 2013, 263, 322-333.

2.                  Results and discussion. I think that the authors made a good characterization of the nanoparticles.I would like to ask if this method is adequate to removal the ions in large quantities of water, i. e. applicable in a waste-water treatment plant and not only at laboratory scale. What is the "real" feasibility also taking into account economical aspects?

Action/response: The magnetic nanoparticle is already known as eco-friendly and used many technological applications. PVA is a hydrophilic and biocompatible polymer, also mostly used for water treatment. The gelatin is a polymer as a mixture of proteins. The combination of these materials are very eco-friendly and can be used especially for biomedical properties.

3.                  Effect of pH. pag 5 line 136. It is not true that the maximum adsorption capacity was found at pH 6 because the authors did not show the results at pH 7. I am aware that at this pH, the precipitation of heavy metal hydroxide occurs. I just suggest to rewrite the text to clarify this aspect.

Action/response: Section 2, Page 5: The following sentences were added into result and marked red:

If the pH more than 6, the heavy metal ions will be precipitated from the aqueous solution [33].

[33] Sutirman Z.A., Sanagi M. M., Karim K. J. A., Ibrahim W. A. W., Jume B. H., Equilibrium, kinetic and mechanism studies of Cu(II) and Cd(II) ions adsorption by modified chitosan bead, Int J Biol Macromol, 2018, 116, 255-263.

4.                  Adsorption kinetics, pag 6, line 170. "...functional groups such as carboxyl, amino, and hydroxyl groups on the surfaces of both adsorbents, ..., were active in heavy metal ion bindin..." How is it demonstrated?

Action/response: The functional groups of Fe3O4/gel and Fe3O4/gel/PVA such as carboxyl, amino, and hydroxyl groups were indicated by FTIR analysis, which can be chelating with the heavy metal ions. The results of FTIR were demonstrated in revision manuscript clearly.

5.                  Table 1. I don't think it would be necessary to include the sorption kinetic parameters corresponding to pseudo-first order since this model can not describe the results.

Action/response: The adsorption kinetics are described pseudo-first-order and pseudo-second-order models, while these indicate the physical or chemical adsorption process. In Table 1, the correlation coefficient R2 of both models are near 1, which means the adsorption processes are fitting well to both models. So we discussed about them here.

6.                  Experimental. Regeneration. How do these acidic solution enriched with ions should be managed as a residue? Please, specify.

Action/response: Section 4, page 15: “Copper and Zinc are very important metals for industries, electroplating, paint, pigments, wood, metal plating, and so on. These enriched acidic solutions can be concentrated and continue used to produce their salts and materials.” was added into the conclusion part.

Round 2

Reviewer 1 Report

The authors have improved their manuscript with their revision. They improved the description of experimental details for TGA, XRD and IR measurements. Furthermore, the authors improved referencing and discussing results throughout their manuscript. DLS measurements have been added which confirm the TEM results. The authors conducted new XRD Furthermore, the theory of adsorption theory has been moved to the experimental/theory section facilitating an easier readability of the manuscript.

However, I still have some major concerns with the manuscripts which should be addressed by the authors:

1.       The nomenclature of magnetic particles should be changed from Fe3O4 to iron oxide nanoparticles or superparamagnetic iron oxide nanoparticles. As the authors state, they observe reflections corresponding to maghemite and I agree that their nanoparticles might be a mixture of magnetite and maghemite nanoparticles or even particles with mixed oxidation states, it is just wrong to name them Fe3O4. Even if other authors might wrongly name these kind of particles magnetite, these particles are no magnetite (Fe3O4) and I will not recommend publication of this manuscript until this nomenclature changed. If you call it Fe3O4 you deliberately spread a wrong nomenclature of these particles and this is in no way compatible with a scientific code of conduct.

2.       FTIR spectroscopy: Your graphs are still labelled wrongly. The Y-axis is labelled transmission in % but you show different spectra with an offset. Please change the labelling of you y-axis to arbitrary units and state the manipulations/normalizations of your spectra in the experimental part. Also add a sentence to the experimental parameters of your IR measurements explaining the measuring method (Transmission). The discussion of the IR section is improved, even though I am still a bit skeptical on the visibility of amide bands. In my opinion, you can also state that the binding of gelatin and PVA is not easy to observe from the spectra. From transmission IR it is really difficult to observe particle coating. I believe you that the particles are coated but you do not have to exaggerate your IR results. You should add the statement, that the coating cannot be observed and only small signals corresponding to amide bands and C-O vibrations can be observed to your discussion. Furthermore, there is no OH vibration around 1350 cm-1. Please erase the sentence containing this statement.

3.       Thank you for adding your DLS measurements. Please add the experimental details in the corresponding section as well.

4.       Magnetization curves: Please add the measured data points in your graphs. You can keep the fitted lines as well but you should add the data points. Another possibility would be to exactly state the number and X-axis positions of measured points in your experimental details.

5.       Adsorption isotherms: I completely understand, that high initial concentrations of copper and zinc might not be interesting for industrial applications. However, in order to explain the binding behavior, at least one data point (1 mg/mL) per coating would                                                                     improve your models. If there are issues with the precipitation of metal salts, I think you can just add at least two data points within your actual investigated range. I think 7 or 8 data points are much better to explain your three different adsorption models than 5 data points. Five data points are just too few to really be able to distinguish between different adsorption models.

Final remarks: I really like your approach and how you are comparing these three adsorption models for heavy metal ion adsorption. I just want to help you to get an interesting paper published which contains experimental descriptions which can be repeated by others as well as results and discussions which are scientifically sound.

Author Response

Response to Reviewer’s Comments on Molecules-379597 “Adsorption of Cu(II) and Zn(II) ions from aqueous solution by gel/PVA-modified magnetic nanoparticles” By: Anudari Dolgormaa1, Chang-jiang Lv1, Yin Li1, Jian Yang2, Jun-xing Yang3, Peng Chen1, Hong-peng Wang1,*, Jun Huang1,* We appreciate very much the constructive and encouraging comments made by the reviewer. This manuscript was revised by considering all comments and suggestions carefully. Detailed responses to each comment and a summary of revisions made are given below. Reviewer 1# The authors have improved their manuscript with their revision. They improved the description of experimental details for TGA, XRD and IR measurements. Furthermore, the authors improved referencing and discussing results throughout their manuscript. DLS measurements have been added which confirm the TEM results. The authors conducted new XRD. Furthermore, the theory of adsorption theory has been moved to the experimental/theory section facilitating an easier readability of the manuscript. However, I still have some major concerns with the manuscripts which should be addressed by the authors: 1. The nomenclature of magnetic particles should be changed from Fe3O4 to iron oxide nanoparticles or superparamagnetic iron oxide nanoparticles. As the authors state, they observe reflections corresponding to maghemite and I agree that their nanoparticles might be a mixture of magnetite and maghemite nanoparticles or even particles with mixed oxidation states, it is just wrong to name them Fe3O4. Even if other authors might wrongly name these kind of particles magnetite, these particles are no magnetite (Fe3O4) and I will not recommend publication of this manuscript until this nomenclature changed. If you call it Fe3O4 you deliberately spread a wrong nomenclature of these particles and this is in no way compatible with a scientific code of conduct. Action/response: The title of manuscript was changed. Adsorption of Cu(II) and Zn(II) ions from aqueous solution by gel/PVA-modified super-paramagnetic iron oxide nanoparticles. Furthermore, the nomenclature of magnetic particles was changed from Fe3O4 to super-paramagnetic iron oxide nanoparticles (SPIONs) in the manuscript. 2. FTIR spectroscopy: Your graphs are still labelled wrongly. The Y-axis is labelled transmission in % but you show different spectra with an offset. Please change the labelling of you y-axis to arbitrary units and state the manipulations/normalizations of your spectra in the experimental part. Also add a sentence to the experimental parameters of your IR measurements explaining the measuring method (Transmission). The discussion of the IR section is improved, even though I am still a bit skeptical on the visibility of amide bands. In my opinion, you can also state that the binding of gelatin and PVA is not easy to observe from the spectra. From transmission IR it is really difficult to observe particle coating. I believe you that the particles are coated but you do not have to exaggerate your IR results. You should add the statement, that the coating cannot be observed and only small signals corresponding to amide bands and C-O vibrations can be observed to your discussion. Furthermore, there is no -OH vibration around 1350 cm-1. Please erase the sentence containing this statement. Action/response: The FTIR figure showed the five different materials spectra. The databases of the materials were plotted by multi curve from Origin Pro 9.1 software, and after plot did not use the manipulation and normalization. Each spectra of the samples were one normal spectra and then they included in one figure. Section 3, Page 11: The sample preparation for FTIR spectroscopy added into the manuscript. The fine powder of KBr was mixed with the sample (0.1-0.5%) and then ground to get homogenous mixture. The homogenous mixture was compressed under 10 MPa pressure to a small disk for analysis (Transmission ~10%). The data of the sample from FTIR were plotted by Origin Pro 9.1 software. Section 2, Page 2: The stretching vibration of –OH (around 1350 cm-1) was erased from the result and discussion part of the FTIR paragraph. Section 2, Page 2: Moreover, as can be seen from FTIR results of SPIONs/gel and SPIONs/gel/PVA, the PVA and gelatin layers cannot be observed easily because of the gelatin and PVA were showed similar functional groups in their spectra, and small signals of the amide bands and C-O stretching vibrations were observed from the SPIONs/gel and SPIONs/gel/PVA. “added into the discussion of FTIR result. Section 2, Page3: The Figure 1 FTIR spectra of (a) PVA, gelatin, (b) SPIONs, SPIONs/gel and SPIONs/gel/PVA. was changed. (a) (b) 3. Thank you for adding your DLS measurements. Please add the experimental details in the corresponding section as well. Action/response: Section 3, Page 10, the DLS measurement performance added into the characterization of adsorbent. The sizes of the Fe3O4, Fe3O4/gel, and Fe3O4/gel/PVA were measured by Dynamic Light Scattering (DLS). All DLS measurements were performed with a Malvern Zetasizer Nano ZS particle analyzer (ZEN3600, Malvern, United Kingdom) at wavelength of 633 nm and He-Ne laser at scattering angle of 173° at 25 °C. In this analysis, 1 mL of particle suspension (each sample was diluted to approximately 0.1 mg/mL) was employed and placed in 10 mm x 10 mm quartz cuvette. The intensity and average diameters were calculated by the Zetasizer Nano Software 7.01. 4. Magnetization curves: Please add the measured data points in your graphs. You can keep the fitted lines as well but you should add the data points. Another possibility would be to exactly state the number and X-axis positions of measured points in your experimental details. Action/response: Section 2, Figure 4. Magnetization curve of Fe3O4/gel and Fe3O4/gel/PVA were changed. 5. Adsorption isotherms: I completely understand, that high initial concentrations of copper and zinc might not be interesting for industrial applications. However, in order to explain the binding behavior, at least one data point (1 mg/mL) per coating would improve your models. If there are issues with the precipitation of metal salts, I think you can just add at least two data points within your actual investigated range. I think 7 or 8 data points are much better to explain your three different adsorption models than 5 data points. Five data points are just too few to really be able to distinguish between different adsorption models. Final remarks: I really like your approach and how you are comparing these three adsorption models for heavy metal ion adsorption. I just want to help you to get an interesting paper published which contains experimental descriptions which can be repeated by others as well as results and discussions which are scientifically sound. Action/response: Section 2, Page 8: The two data points of adsorption of Cu(II) and Zn(II) ions on SPIONs/gel and SPIONs/gel/PVA (initial concentration 0.75 and 1 mg/mL) were measured and added into the result and discussion part of the adsorption isotherm and the experimental part. Furthermore, the adsorption isotherm databases were also changed a bit when the adsorption data fitted by Langmuir, Freundlich, and Sips models. Section 2, Page 8, Line 187: “The adsorption capacity of Cu(II) on bare iron oxide was just 0.15 mg/g, much less than that on another two adsorbents (15-25 mg/g), so that the following adsorption experiments were carried on SPIONs/gel and SPIONs/gel/PVA.” was changed to Section 2, Page 8, Line 190. Page 1, Abstract part was changed Super-paramagnetic iron oxide nanoparticles (SPIONs)/gelatin (gel)/polyvinyl alcohol (PVA) nanoparticles were designed and synthesized by the co-precipitation method and further modified with gel and PVA. These nanoparticles were used for the removal of Cu(II) and Zn(II) from aqueous solutions. The adsorbents were rich in different functional groups for chemisorption and showed effective adsorption properties. The adsorption of Cu(II) and Zn(II) on the SPIONs/gel and SPIONs/gel/PVA materials were investigated with respect to pH, adsorption kinetics, and adsorption isotherms. The adsorption data was fitted to the Langmuir, Freundlich, and Sips models at the optimum pH 5.2 (±0.2) over 60 min; SPIONs/gel showed maximum adsorption capacities of 47.594 mg/g and 40.559 mg/g for Cu(II) and Zn(II); SPIONs/gel/PVA showed those of 56.051 mg/g and 40.865 mg/g, respectively. The experimental data fitted the pseudo-second-order model, indicating that the process followed chemical monolayer adsorption. In addition, the SPIONs/gel/PVA showed better stability and Cu(II) adsorption efficiency than SPIONs/gel. The adsorption isotherm Figure 8 were changed. Figure 8. Equilibrium adsorption isotherm of Cu(II) (a) and Zn(II) (b) by the SPIONs/gel and SPIONs/gel/PVA.   The Table 2 fitted results of the adsorption isotherm of the Cu(II) and Zn(II) by SPIONs/gel and SPIONs/gel/PVA were changed. Isotherm Parameters Cu (II) Zn (II) SPIONs/gel SPIONs/gel/PVA SPIONs/gel SPIONs/gel/PVA Langmuir model Qmax (mg/g) 47.594 56.051 40.559 40.865 KL (mL/mg) 6.66 3.397 5.727 3.267 R2 0.995 0.993 0.997 0.999 Freundlich model KF ((mg/g)(mL/mg)1/n) 43.991 45.658 36.515 32.866 1/n 0.3164 0.4373 0.3367 0.4399 R2 0.993 0.976 0.986 0.979 Sips model Qmax (mg/g) 60.422 54.823 43.974 38.976 KS (mL/mg)1/n 2.382 3.713 3.854 3.961 1/n 0.6816 1.0319 0.8632 1.071 R2 0.999 0.991 0.998 0.999 Section 2, Page 9: The dimensionless separation parameter RL value from the Langmuir isotherm model, the 1/n value from Freundlich model and the maximum adsorption capacity of Cu(II) and Zn(II) on SPIONs/gel and SPIONs/gel/PVA from Langmuir model were changed. where C0 is initial concentration of the metal ions (mg/mL) and the definition of KL is as explained for equation 4. The RL value is classified as irreversible (RL = 0), linear (RL = 1), favorable (0 < RL < 1) and unfavorable (RL > 1) [39, 40]. In this work, all the values of RL are between 0.1305 and 0.2274, indicating that the experimental data fall between zero and 1, which is an indication of the favorable adsorption of the two metal ions on the adsorbents. The 1/n values from the Freundlich model for the adsorption of SPIONs/gel were 0.3164 and 0.3367 for Cu(II) and Zn(II); SPIONs/gel/PVA were those of 0.4373 and 0.4399, respectively. These values are lower than 1, indicating that the adsorption of Cu(II) and Zn(II) is favorable on the both adsorbents. Section 3, Page 12: The initial concentration of two metal ions added into the experimental part of adsorption isotherm. 25 mL of Cu (II) and Zn (II) metal ions solutions with different initial concentrations of 0.1, 0.2, 0.3, 0.4, 0.5, 0.75 and 1 mg/L was mixed with 0.025 g of adsorbents and shaken at 25 °C and 130 rpm. Section 2, Page 18: Result and discussion of the regeneration part. The adsorption capacity of Cu(II) on SPIONs/gel and SPIONs/gel/PVA were decreased by 5.05% and 7.28%. The adsorption capacity of Zn(II) on SPIONs/gel and SPIONs/gel/PVA were decreased by 7.87% and 12.57%, after three cycles, respectively. The reusability of these materials were compared with some magnetic adsorbent materials, which was used to similar preparation of iron oxide nanoparticles, initial concentration of heavy metal ions and cycling usage conditions. The adsorption capacity of Cu(II) on the Mg/Fe layered double hydroxide loaded with Magnetic (Fe3O4) carbon spheres [48], and the magnetic Fe3O4/carbon nanotube (CNT) [49] were decreased by 18.6%, and 17.6%, after three cycles usage, respectively. The adsorption capacity of Cu(II) and Zn(II) on magnetic nanoparticles decorated tea waste (Fe3O4-tea) was decreased by 9.5% and 17.1% after six cycles [46]. As shown in Figure 9, after three cycles that the SPIONs/gel and SPIONs/gel/PVA exhibit good regeneration performance and can support long-term usage in the removal of Cu(II) and Zn(II).
